# Chronic Electrical Stimulation of the Superior Laryngeal Nerve in the Rat: A Potential Therapeutic Approach for Postmenopausal Osteoporosis

**DOI:** 10.3390/biomedicines8090369

**Published:** 2020-09-22

**Authors:** Kaori Iimura, Nobuhiro Watanabe, Philip Milliken, Yee-Hsee Hsieh, Stephen J. Lewis, Arun Sridhar, Harumi Hotta

**Affiliations:** 1Department of Autonomic Neuroscience, Tokyo Metropolitan Institute of Gerontology, Tokyo 173-0015, Japan; kiimura@tmig.or.jp (K.I.); watanobu@tmig.or.jp (N.W.); 2Galvani Bioelectronics, Stevenage SG1 2NY, UK; philip.h.milliken@galvani.bio (P.M.); arun.x.sridhar@galvani.bio (A.S.); 3Division of Pulmonary, Critical Care and Sleep Medicine, Department of Medicine, University Hospital Cleveland Medical Center, Case Western Reserve University, Cleveland, OH 44106, USA; yxh13@case.edu; 4Division of Pulmonology, Allergy and Immunology, Department of Pediatrics, School of Medicine, Case Western Reserve University, Cleveland, OH 44106, USA; sjl78@case.edu

**Keywords:** superior laryngeal nerve, calcitonin, thyroid gland, osteoporosis, bone mineral density, neuromodulation, electrical stimulation, ovariectomy

## Abstract

Electrical stimulation of myelinated afferent fibers of the superior laryngeal nerve (SLN) facilitates calcitonin secretion from the thyroid gland in anesthetized rats. In this study, we aimed to quantify the electrical SLN stimulation-induced systemic calcitonin release in conscious rats and to then clarify effects of chronic SLN stimulation on bone mineral density (BMD) in a rat ovariectomized disease model of osteoporosis. Cuff electrodes were implanted bilaterally on SLNs and after two weeks recovery were stimulated (0.5 ms, 90 microampere) repetitively at 40 Hz for 8 min. Immunoreactive calcitonin release was initially measured and quantified in systemic venous blood plasma samples from conscious healthy rats. For chronic SLN stimulation, stimuli were applied intermittently for 3–4 weeks, starting at five weeks after ovariectomy (OVX). After the end of the stimulation period, BMD of the femur and tibia was measured. SLN stimulation increased plasma immunoreactive calcitonin concentration by 13.3 ± 17.3 pg/mL (mean ± SD). BMD in proximal metaphysis of tibia (*p* = 0.0324) and in distal metaphysis of femur (*p* = 0.0510) in chronically SLN-stimulated rats was 4–5% higher than that in sham rats. Our findings demonstrate chronic electrical stimulation of the SLNs produced enhanced calcitonin release from the thyroid gland and partially improved bone loss in OVX rats.

## 1. Introduction

Osteoporosis is characterized by reduced bone mineral density (BMD) and changes in bone morphometry, which increases the risk of fracture. Osteoporosis is one of the most common health problems of the elderly, estimated to affect 200 million women worldwide. Drug therapy is mainly used for the treatment of osteoporosis, but the current pharmacotherapy options have various limitations such as side effects (e.g., nausea or pain with bisphosphonates) and treatment compliance [1,2].

Calcitonin (CT), a peptide hormone secreted from the thyroid gland, suppresses osteoclast activity, preventing bone loss [3]. Exogenous CT has been widely used as a bone resorption inhibitor for the treatment of osteoporosis. However, CT products have a short half-life and produce large amounts of antibodies against CT, so it is necessary to intermittently administer the minimum effective dose [4]. Patients have to visit the clinic frequently and receive CT injections, increasing patient burden for such treatment regimens, especially for active and busy patients. Therefore, the development of alternative therapies to replace CT drug administration would improve the limitations of current osteoporosis treatment, where bioelectric medicine offers a novel approach. If bioelectrical neuromodulation increases secretion of endogenous CT, there will be no risk of antibody production or patient compliance. Therefore, we examined the effects of electrical stimulation of peripheral nerves on CT secretion from thyroid gland.

We recently showed in anesthetized and artificially ventilated rats that electrical stimulation of the superior laryngeal nerve (SLN) innervating the thyroid gland increases immunoreactive CT (iCT) secretion into thyroid venous blood [5]. In this study, firstly to examine the effect of efferent nerve stimulation, the SLN was cut and supramaximal electrical stimulation applied to the cut peripheral end of the nerve, with iCT secretion being increased at 40 Hz. SLN includes myelinated (mostly afferent) nerves in addition to unmyelinated autonomic efferent nerves. Selective stimulation of myelinated nerve fibers of intact SLNs was then tested with low intensity currents (0.5 ms, 4–60 μA) at 40 Hz, also producing comparable increases in iCT secretion. These experiments demonstrated that both efferent and selective afferent nerve stimulation of SLN had similar effects on iCT secretion, suggesting that myelinated afferent fibers of SLN have a role in facilitating CT secretion from the thyroid gland through reflex excitation of the parasympathetic efferent nerve fibers to the thyroid gland.

The purpose of this study was to progress from acute stimulation in anaesthetized animals and to establish a conscious experimental animal model to translate results obtained in anesthetized animals for potential clinical application in osteoporosis. The SLN is known to include afferent nerve fibers that can inhibit respiration [6,7,8]. Therefore, it is necessary to identify parameters that do not adversely affect respiratory measures by SLN stimulation.

First, we investigated whether SLN electrical stimulation increases systemic iCT without affecting respiration in conscious rats. Second, to clarify the potential of SLN stimulation to improve osteoporosis, we investigated the effects of chronic SLN stimulation on BMD and bone histomorphometry in the ovariectomized (OVX) rat model of osteoporosis [9,10].

## 2. Materials and Methods

The study comprised two arms (shown in Table 1). Healthy control animals (*n* = 16) were initially used to examine the effects of SLN stimulation on respiration and systemic CT concentrations. Subsequently, selected stimulation parameters from these studies were progressed into a chronic OVX model of osteoporosis for bone analysis, compared to sham (non-SLN stimulated) OVX animals (*n* = 11). In one of the 11 rats, we found inflammation around a cuff at study end, so that the rat was excluded from the analysis. Experiments were conducted at Tokyo Metropolitan Institute of Gerontology (TMIG), except for four rats in the respiration analysis that was carried out at Case Western Reserve University (CWRU). All experiments conducted at TMIG were in accordance with Guidelines for the Appropriate Implementation of Animal Experiments (established by the Japan Scientific Council in 2006) and approved by the Animal Care and Use Committee of TMIG (approved on 23 August 2019, approval number 18026-2). For the respiration assessment at CWRU, all experimental procedures were carried out in accordance with the National Institutes of Health Guide for the Care and Use of Laboratory Animals and the CWRU Institutional Animal Care and Use Committee (approved on 19 March 2019, approval number 2016-0034). Animals at TMIG were housed in individually ventilated caging with paper bedding under a specific pathogen-free condition. Environmental conditions at TMIG were kept stable (22 ± 1 °C of temperature, 55 ± 5% of relative humidity) with a 12-h light, 12-h dark cycle (light on at 8 a.m., light off at 8 p.m.). Foods and water (filtered tap water with 2 ppm of chlorine) were provided ad libitum except during respiratory data recording. Animals were kept in a conventional environment after surgery at CWRU.

### 2.1. Effect of SLN Stimulation on Respiration and iCT Levels in Healthy, Conscious Rats

#### 2.1.1. Surgery Overview

Nine Sprague-Dawley and seven Wistar male rats (weight range 325–475 g) were used. Under isoflurane anesthesia (1–4% inhalation), survival surgery was performed with aseptic procedures. Procaine hydrochloride was subcutaneously administered to the incision site in advance for local anesthesia. The skin was longitudinally incised at midline. To stimulate SLNs, cuff electrodes were implanted on bilateral SLNs, and a head socket connected with cuff electrodes via their lead wires was fixed on the skull. For blood sampling (in 10 rats), a catheter was implanted in the superior vena cava which was connected to a vascular access button fixed on the back skin and muscle. At the end of surgical procedures, warmed saline (5 mL of total volume) was subcutaneously (s.c.) administered for hydration. Further procedural details are provided below.

##### Implantation of Intravenous Catheter

A polyurethane catheter with a round tip (outer diameter: 1.02 mm, inner diameter: 0.62 mm, total length: 15 cm; C30PU-RJV1303, Instech Laboratories Inc., Plymouth Meeting, PA, USA) was used to minimize damage of venous endothelium. The other tip of the catheter was connected to a vascular access button (VABM1B/22 or VABM2B/22R22, Instech) covered with a protective metal cap (VABM1C or VABRC, Instech). Firstly, the vascular access button connected to the catheter was sewn to the back skin and muscle with a sterile 4-0 silk suture. Secondly, the other catheter end was passed subcutaneously from the back to the front neck that had been incised. The right jugular vein was isolated from other tissues, and the catheter was inserted until the tip was located in the superior vena cava and then fixed with a sterile 6-0 silk suture. The skin was sewn with sterile 4-0 nylon suture and fixed with a tissue adhesive (3M^TM^ Vetbond^TM^ Tissue Adhesive, 3M, St. Paul, MN, USA). The catheter was prefilled with heparin glycerol (Catheter Loc Solution, Braintree Scientific, Inc., Braintree, MA, USA). By replacing the heparin glycerol in the catheter every four days, the efficacy of the catheter was successfully maintained for two weeks.

##### Implantation of Cuff Electrode on the SLN

Cuff electrodes (inner diameter: 200 μm; length: 2 mm; AirRay research Micro Cuff Tunnel 1041.2407.52, CorTec GmbH, Freiburg, Germany) were implanted on left and right SLNs. Before implantation, the cuff electrode was connected to a head socket (MS303-120 or M363, PlasticsOne, Roanoke, VA, USA) via lead wires (length: 7–15 mm). Firstly, the head socket was fixed to the skull with three mounted screws, high-strength adhesive, and dental cement. Then, the cuff and lead wires connected were passed subcutaneously, using a trocar, to the skin incision in the front neck. Secondly, cuffs were placed on the SLNs of the anterior cervical region approximately 8 mm lateral from the thyroid (as shown in Figure 1C of our previous study [5]). To expose the SLN, the ventral neck muscles were separated without transection. Special care was taken to minimize damage of SLNs and surrounding tissues including blood vessels. The cuff electrode and the nerve were fixed with fibrin sealant (TISSEEL, Baxter, Deerfield, IL, USA). The head socket and dental cement were covered with sutured scalp. The exteriorized part of the head socket was protected with a cap (303DCA or 363DC, PlasticsOne). After each experiment, the position of the electrode was checked postmortem, to determine whether the cuff was fixed on left and/or right SLN.

##### Prevention of Infection and Postoperative Care

On the day of surgery, an antibiotic (cefazoline 50 mg/kg, s.c.) and analgesics (carprofen 5 mg/kg, s.c. and buprenorphine 0.03 mg/kg, s.c.) were administered under isoflurane anesthesia before starting surgery. An ointment (Puralube Ophthalmic Ointment, Dechra Veterinary Products, Overland Park, KS, USA) was applied to the eyes for preventing dry eye. The skin of the surgical site was shaved and sterilized with skin cleanser (Isodine^®^, Shionogi Co., Ltd., Osaka, Japan or ChlorHex-Q scrub, Vedco Inc., Saint Joseph, MO, USA) and 70% isopropyl ethanol, while keeping body temperature by a heating mat (T-care, Kimuramed, Tokyo, Japan). After completing the surgery, the anesthesia was disconnected. Observing complete awakening from the anesthesia, we housed rats individually in an animal facility. An antibiotic (cephazoline, 50 mg/kg, s.c., once/day) and analgesic anti-inflammatory drug (carprofen, 5 mg/kg, s.c., once/day) was administered for two days and five days after surgery, respectively. For a week after surgery, soft foods, such as pellets softened with drinking water, steamed rice, and high-nutrition jelly (DietGel 76A, DietGel Recovery, ClearH_2_O, Portland, ME, USA), and oral rehydration solution (Rehydration Support, Royal Canin, Gard, France) were put on the floor in the animal cage, in addition to normal food (solid pellets, CRF-1, Oriental Yeast Co., Tokyo, Japan).

#### 2.1.2. Acute Conscious Stimulation of the SLN

Following a recovery period of two weeks after surgery, rats were temporarily anesthetized with 1.5% isoflurane and a tether protected by a metal spring (305-SL/2 or 363-419/6, PlasticsOne) was connected to the head socket. A blood collection catheter (length: 40 cm, KVABM1T/25, Instech) was connected to the vascular access button on the day of blood sampling. Animals were placed in an acrylic chamber (30 × 30 × 35 cm for blood sampling, approximately 20 cm in diameter and 10 cm in height for respiration analysis) and anesthesia was disconnected. After complete recovery from anesthesia based on the recovery of the animal’s behavior and locomotion, the SLN was stimulated electrically using the same method described in the preceding paper [5] with some modifications. Rectangular pulses (duration: 0.5 ms) were used to stimulate myelinated fibers of the SLN at 40 Hz (SEN-8203 with SS-203J, Nihon Kohden, Tokyo, Japan, or S48, Grass Technologies Co., West Warwick, RI, USA). The current intensity was increased (15–120 μA) for respiratory analysis experiments and fixed at 90 μA for blood sampling experiments. Animals in the sham group of blood sampling experiment were also tethered, but electrical stimulation was not applied to the SLNs.

#### 2.1.3. Respiratory Analysis

A whole-body plethysmograph (Buxco Research Systems Co., Wilmington, NC, USA, or Emka Technologies, Paris, France) was used to measure tidal volume and ventilation rate in conscious rats to test the effects of SLN stimulation in seven rats. The measurement was performed following acclimatization to the chamber for about 1 h.

#### 2.1.4. Blood Sampling and iCT Measurement

Blood was collected while stimulating the SLN in conscious rats. Prior to blood sampling, 0.5 mL of heparinized saline (200 U/mL of heparin) and 2 mL of 4% Ficoll PM70 were administered in divided doses over 1 h. A total of nine blood samples were then intermittently collected from each rat via the venous catheter. Each sample, consisting 0.5 mL of blood, was collected using a syringe over a 6-min period. After the end of each blood sampling, the same volume (0.5 mL) of fluid replacement (mixture of 5.4 mL of 4% Ficoll PM70 solution and 0.4 mL of heparinized saline (200 U/mL)) was administered via the same catheter. Samples were collected 20 min before, during, and 40 min after SLN stimulation. In a pilot study in which stimulation was repeated at 60-min intervals, there was a tendency for the next stimulation to be applied before the increased iCT level returned to its prestimulus baseline levels. Therefore, the stimulation interval was increased to 120 min in this study. Blood samples were processed, as described previously [5], measuring plasma sample iCT concentrations using an ELISA kit (rat CT ELISA kit, MBS703165, MyBioSource, San Diego, CA, USA, minimum detectable dose: 0.39 pg/mL, intra-assay precision: Coefficient of variation % < 8%, inter-assay precision: Coefficient of variation % < 10%).

### 2.2. Effect of SLN Stimulation on Osteoporosis in OVX Rats

#### 2.2.1. Time Schedule

Figure 1 shows the time schedule of Experiment 2. Ten Sprague-Dawley female rats were used. Rats received OVX surgery at 12 weeks of age by a specialist at Japan SLC, Inc., Shizuoka, Japan. They were delivered to the animal facility of TMIG at a week after OVX. After two weeks of acclimatization, they received aseptic surgery of cuff implantation and head socket (MS363 with E363/0, PlasticsOne) fixation for SLN stimulation, as described above (Section 2.1.1). After a recovery period of two weeks following implantation surgery, SLN stimulation or sham control nonstimulation was applied for 3–4 weeks (i.e., from the 5th week to the 8th–9th week after OVX). This duration (3–4 weeks) was based on the minimum duration of studies examining the effect of exogenous CT administration in OVX rats [11,12]. The duration was dependent on electrode functionality. On the day of euthanasia, body weight was measured and femurs and tibias were dissected. Animals were fed pellet diet containing 0.98% calcium (Labo MR Stock, Nosan Corporation, Yokohama, Japan).

#### 2.2.2. Tethering

In order to perform SLN electrical stimulation repeatedly for 3–4 weeks, the animals were kept tethered during that period. A tether protected by a metal spring (length: 40 cm, 363-363, PlasticsOne) was connected to the head socket one day before the onset of stimulation for acclimation. The tether in the cage was 40 cm long and connected to a rotatable swivel (Miniature slip ring #775, Adafruit Industries LLC, New York, NY, USA) to allow the animal to move freely. A red box (10 × 16.5 × 10 cm) was placed in the cage (30 × 32 × 18 cm) as an enrichment.

#### 2.2.3. Chronic Stimulation of the SLNs

For chronic stimulation, in order to reduce tissue damage, we used a charge-balanced biphasic pulse of 0.5-ms width with an interphase delay of 5 μs. A tetanic stimulation (with a current intensity of 90 μA at 40 Hz) was based on the respiratory assessment and prior anesthetized work [5]. Stimulation was applied for a period of 8 min (sufficient to release maximal levels of CT), starting at 11:00 a.m., repeated every hour, seven times a day, and four days per week. For such scheduled stimulation, a multichannel electrical stimulator (PlexStim™ Electrical Stimulator 2.0, Plexon Inc., Dallas, TX, USA) was controlled by an external trigger (Power3 1401, Cambridge Electronic Design Limited, Cambridge, UK).

On the day of stimulation, the functionality of each electrode was confirmed by observing induction of swallowing reflex during SLN stimulation and/or the stimulus waveform on the oscilloscope. Additionally, the impedance of each electrode was measured once a week. An acceptable range was less than 50 kΩ, measured at 100 mV, 10 kHz (Minirator MR-PRO, NTi Audio AG, Schaan, Liechtenstein).

Following each experiment at termination, the position of each cuff was assessed postmortem, to determine whether the cuff remained positioned on left and/or right SLN. Simultaneously, the condition of the tissue around the cuff was checked under binocular microscope. Condition of tissue was normal for all rats used for analysis.

#### 2.2.4. Ex Vivo Bone Assessments

Rats were euthanized at 8–9 weeks after OVX (when rats were 20–21 weeks old) by an overdose of pentobarbital. The femur and tibia were dissected bilaterally. Bones were immersed in 70% ethanol for bone assessments, including BMD measured by dual energy X-ray absorption (DXA) and bone histomorphometry. All analysis was performed by a specialist (Kureha Special Laboratory Co. Ltd., Tokyo, Japan) in a blinded manner.

DXA: The right tibia and left femur immersed in 70% ethanol were dried, and the BMD and bone mineral content were measured using DXA. The whole bone was divided into 20 parts of equal length to quantify each regional value to eliminate the influence of inter-individual difference in bone length [13].

Bone histomorphometry: Nondecalcified bone specimens were made in the proximal tibia (left side), and the secondary cancellous bone beneath the growth plate in one frontal section for each sample was analyzed. Bone structure (cancellous bone volume, trabecular thickness, and number of trabeculae), bone formation (osteoid surface, osteoblast surface, bone calcified surface), and bone resorption (erosional surface, osteoclast surface, and the number of osteoclasts) were determined.

To determine the bone calcification rate, fluorescent calcium chelate dye (calcein, 0.8% solution, 1 mL/kg) was injected subcutaneously at 6–8 days and one day before euthanasia.

### 2.3. Statistical Analysis

Statistical analyses were performed using a statistics software (Prism ver. 6, GraphPad software Inc., San Diego, CA, USA). Values are expressed as the mean ± SD or median and interquartile range (25%–75%). The respiratory function data were analyzed by a Friedman’s test. For CT and bone analysis, values in the stimulation group were compared with the sham stimulation group using Student’s *t*-test or Mann–Whitney U-test. One-tailed tests were used for analysis of our main hypothesis that CT concentration and BMD in the stimulation group would be higher than that in the sham stimulation group. Otherwise, two-tailed tests were used. Statistical significance level was set at 5%.

## 3. Results

### 3.1. Acute SLN Stimulation in Conscious Rats Increased Systemic iCT without Affecting Spontaneous Respiration (Experiment 1)

Electrical pulses of 0.5-ms duration with varying intensities from 15 to 120 μA, at a frequency of 40 Hz, were continuously applied to the bilateral (in five rats) or unilateral (in two rats) SLN. In addition to the continuous SLN stimulation, electrical pulses at 90 μA of intensity were applied intermittently with 1 s on/4 s off and with 1 s on/8 s off. SLN stimulation using these parameters did not cause any adverse behavior in conscious rats. However, a swallowing-reflex response was observed during intermittent and/or continuous 90 μA stimulation in six of seven rats. SLN continuous stimulation with an intensity from 15 μA to 120 μA did not affect the ventilation rate or tidal volume in all seven rats in which stimulation was applied for a period of either 30 s or 5 min. Figure 2 summarizes the respiratory function data from seven rats during continuous SLN stimulation with 90 μA. SLN stimulation did not affect the ventilation rate, tidal volume, or clinical observations at these levels and was considered safe for use in further conscious rat studies and disease model use.

We then examined the effect of SLN stimulation on systemic iCT and sampled systemic venous blood. In this experiment, an intensity of 90 μA was applied to bilateral (in five rats) or unilateral (in two rats) SLN for 8 min. SLN was stimulated in three different modes as described above, in a randomized order, at an interval of 120 min. In six of the seven rats receiving SLN stimulation (regardless of bilateral or unilateral stimulation), systemic iCT concentration increased during continuous 40-Hz stimulation. The iCT levels recovered to baseline prestimulation levels between 40 to 100 min after the end of stimulation. Figure 3 shows the magnitude of changes from basal control level in iCT concentration in venous plasma during sham stimulation (*n* = 9 in three animals) and continuous SLN stimulation (*n* = 7 in seven animals). Sham animals had undergone SLN electrode implantation, but the SLNs did not receive electrical stimulation. The iCT concentration did not change during sham stimulation, whereas iCT increased by 13.3 ± 17.3 pg/mL (mean ± SD) during SLN stimulation (*p* = 0.0377 compared to sham stimulation). In the case of intermittent stimulation, iCT increase was observed in four of seven animals (1 s on/4 s off) and in three of seven animals (1 s on/8 s off); however, iCT decrease was observed in the other animals so that mean iCT levels did not change during stimulation.

### 3.2. Chronic SLN Stimulation in Conscious Rats Partially Improved Bone Loss Following OVX (Experiment 2)

In preliminary experiments, we used OVX (*n* = 2) and sham OVX (*n* = 2) rats to confirm development of osteoporosis following ovariectomy (OVX) in our experimental conditions, including recovery from head surgery and neck incision. Consistent with previous reports [9,10], in OVX rats body weight was higher (Appendix A) but BMD in the distal femur measured in vivo by radioabsometry was lower (Appendix A) than those in sham OVX rats (same surgical procedure without OVX including skin incision, the exposure and temporary lift of the ovaries, and skin suture). These differences occurred two weeks after OVX and then persisted throughout the observation period. The regional difference of changes in BMD was shown by DXA measure using dissected bones at eight weeks after OVX. The magnitude of reduction was most severe at the metaphysis of distal femur (fourth section from distal end) and that of proximal tibia (third section from proximal end), in which BMD was approximately 70% of sham OVX rats (Appendix A).

To determine whether chronic SLN stimulation (0.5 ms, 90 μA, 40 Hz for 8 min/hour × 7 times/day × 4 days/week) for 3–4 weeks prevents or reduces the progression of osteoporosis development in OVX rats, BMD values in rats with successfully stimulated SLN for 12–16 days (*n* = 5 rats) were compared with those in sham group (*n* = 5, nonstimulated rats). We compared BMD values measured by DXA in the metaphysis of distal femur (fourth section from distal end) and that of proximal tibia (third section from proximal end), where the most significant reduction was observed in OVX rats in preliminary experiments, as outlined above (Appendix A). There was a statistically significant difference between groups in the BMD values, depending on whether SLN was chronically stimulated. The BMD in the distal femur of the stimulation group tended to be higher than that in the sham group (123.8 ± 6.6 vs. 117.5 ± 3.8 mg/cm^2^, mean ± SD, *p* = 0.0510; Figure 4A). The BMD in the proximal tibia in the stimulation group was significantly higher than that in the sham group (112.7 ± 4.0 vs. 108.3 ± 2.4 mg/cm^2^, *p* = 0.0324; Figure 4B). However, even in the stimulation group, the values of BMD were about 40 mg/cm^2^ lower than those in normal rats with ovaries (sham OVX) for both femur and tibia (see values in sham OVX rats in Appendix A).

There were clear differences between OVX rats and sham OVX rats for various other parameters of bone analysis undertaken in the preliminary experiments (particularly, cancellous bone weight, osteoid surface, and osteoclast number) (Appendix A). However, there were no statistically significant findings between OVX SLN stimulation and OVX sham stimulation groups in any of the results on bone histomorphometry (Table 2).

## 4. Discussion

In this study, SLN stimulation (0.5 ms, 90 μA, 40 Hz) was applied to conscious rats two weeks after cuff electrode implantation on the SLN nerves, following surgical and body weight recovery. During SLN stimulation, a swallowing reflex was observed. However, ventilation rate and tidal volume monitored by whole-body plethysmography were unchanged. This result is consistent with a study in unanesthetized rabbits in which SLN stimulation, with intensity at 1.2–1.5 times the threshold for inducing swallowing reflex (0.2 ms, 0.2–0.65 mA, 30 Hz), that did not affect the behavior of the animal [14]. In unanesthetized piglets, when SLNs were stimulated (0.5 ms, 10 Hz) with an intensity near threshold for respiratory changes, which were lower than that for producing discomfort behavior, mean respiratory frequency decreased by 57% during stimulation period of 5 min or 1 h [6,7]. When SLN electrical stimulation was applied to conscious humans, sensation of touch and pain were induced, depending on stimulation parameters [15]. The stimulus we used was high-frequency and low-intensity to selectively stimulate only the large myelinated fibers [5], without additional activation of small-diameter nerve fibers that inhibit respiration or elicit aversive behaviors.

The low-intensity and high-frequency SLN stimulation that we reported to elicit localized iCT secretion from the thyroid gland in anesthetized rats ([5]; using 0.5 ms, 4–60 μA, 40 Hz) also increased systemic iCT concentration by 13.3 pg/mL in conscious rats (this study; using 0.5 ms, 90 μA, 40 Hz). CT, by acute administration, inhibits osteoclast-mediated bone resorption by direct action on osteoclasts: It binds to the CT receptor expressed on the surface of osteoclasts, disrupts the actin ring formed by osteoclasts, and suppresses bone resorption activity. According to Yamauchi et al. [4], bone resorption rate decreases by 1.27% when blood CT increases by 1 pg/mL in humans. If we hypothetically apply this relationship to our rats, the SLN stimulation-induced CT increase is estimated to reduce bone resorption rate by 17%. Although the bone resorption-suppressing effect of CT is immediate and powerful, continuous administration causes unresponsiveness (escape phenomenon). This phenomenon is acquired 48–72 h after administration [16]. In rodents, endogenous CT has also been shown to play a role in regulating postprandial hypercalcemia [17]. A recent study revealed that pharyngeal mechanical stimulation, like during eating, increases iCT secretion by a neurogenic reflex through the SLN [18]. Therefore, stimulation of the SLN during daytime, when rats do not eat, was expected to be effective in inhibiting bone loss. Indeed, the effect of CT administration was optimal in OVX rats when administered during the daytime [19]. Accordingly, in this study we applied SLN stimulation intermittently during daytime in rats after OVX. Intermittent stimulation was utilized, as 8-min stimulation in conscious rats elevated systemic iCT levels for up to 100 min. Therefore, hourly application of SLN stimulation would provide adequate prolonged increases in CT.

Chronic SLN stimulation using this paradigm and duration slightly increased BMD in OVX rats. In this study, 3–4 weeks (8 min × 7 times × 5 days × 3–4 weeks = 14–19 h) of SLN stimulation increased BMD of OVX-induced osteopenia by 4–5% compared to the nonstimulated group. The effect was statistically significant in proximal tibia and nearly significant in distal femur, but was similar in trabecular bone of both tibia and femur. It has been reported that intermittent administration of exogenous CT increased BMD in OVX rats. For example, in rats ovariectomized at 11 weeks old, acute and localized osteopenia was induced in proximal tibia and distal femur within two weeks after OVX, and was prevented or reduced by salmon CT (5 and 20 U/kg, s.c., every two days, for 2–4 weeks) [12]. A smaller dose (equivalent to clinical dosage) of salmon CT (2 U/kg, s.c., every two days, for 60 days) also prevented osteopenia in femur and the fifth lumbar vertebra of OVX rats, measured at 60 days after OVX by DXA [20], a longer time frame than assessed in our study. BMD increase induced by the present protocol of SLN stimulation may be smaller than those induced by exogenous CT in these animal studies [12,20]. However, in clinical studies [21,22], intermittent administration of CT for a longer period (≥6 months) increased BMD of lumbar vertebra in a similar extent to that in the present study. Therefore, 4–5% increase in BMD by only 3–4 weeks of stimulation might be clinically meaningful. Further studies would clarify the optimal protocol of SLN stimulation with a prolonged stimulus duration.

We propose that an increase in CT secretion induced by SLN stimulation, a phenomenon on the basis of the present study, is involved in the inhibition of OVX-induced bone loss. However, a limitation of this study is that we could not show any data directly connecting CT and BMD. Basal CT levels are reportedly lower in postmenopausal women than in premenopausal women and particularly low in osteoporotic patients [23]. In postmenopausal patients with osteoporosis, CT secretion response to Ca^2+^ is attenuated [24]. If the neuronal CT secretory reflex via SLN is maintained in OVX rats, this neural mechanism may be even more important in regulating postmenopausal bone metabolism than in normal conditions. The CT secretion response during SLN stimulation was shown in normal animals in this study. In addition, afferent information from SLN has a reflexive effect not only on thyroid function but also on other autonomic functions. For example, it has been reported that electrical stimulation of SLN reflexively induces vasodilation of the lower extremities of anesthetized animals [25,26]. This response may also contribute to increased BMD in the femur and tibia. Future studies are needed to further elucidate the mechanism of chronic SLN stimulation on bone metabolism and the optimal stimulation paradigm for maximal efficacy in osteoporosis. However, the present findings demonstrate the potential, following successful application of SLN stimulation in conscious animals over several weeks, without any adverse findings or behavioral effects.

In contrast to the slight but significant effect on BMD in proximal tibia, we did not find any significant reduction of osteoclast numbers or bone-resorbing activities or increase in osteogenesis by histological analyses. Currently, our understanding of this unexpected result remains undetermined. One possibility is the difference may be too small to detect through histological examination. Alternatively, the chronic effect on bone cells of endogenous CT might be different from that of exogenous CT, as suggested by recent findings [27]. Salmon CT induces downregulation of CT receptor, resulting in the disappearance of CT effects. Consequently, exogenous CT’s effects on bone is limited in humans, now being used for the treatment of osteoporosis only for the decrease in bone pain via the neurological mechanism [3,28]. However, our initial results with neuromodulation of the SLN suggest increased endogenous CT secretion might operate through mechanisms differing from exogeneous CT for treatment of osteoporosis.

## 5. Conclusions

We found a parameter of SLN stimulation that increased systemic blood CT concentration without affecting spontaneous respiration in conscious rats. Using that parameter, we examined efficacy of SLN stimulation in treating osteoporosis in a rat model of postmenopausal osteoporosis. We found that chronic electrical stimulation of SLN could be applied chronically for several weeks, partially improving BMD in osteoporotic rats. Our initial results suggest that electrical SLN stimulation could provide a novel approach and new bioelectric medicine for treatment of osteoporosis.

## Figures and Tables

**Figure 1 biomedicines-08-00369-f001:**
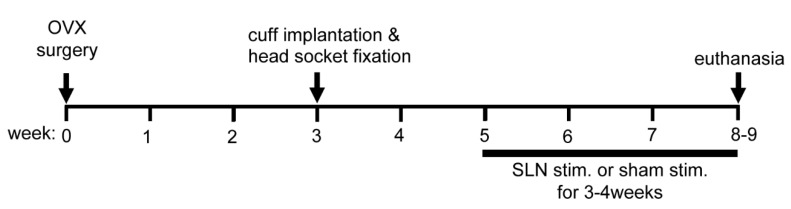
Time schedule of Experiment 2. Rats received ovariectomy (OVX) surgery at 12 weeks of age. They received another surgery of implanting cuff electrodes on the superior laryngeal nerves (SLNs) and fixing a head socket on the skull at three weeks after OVX. SLNs stimulation was started at five weeks after OVX and continued for a period of 3–4 weeks (8 min/h × 7 times/day × 4 days/week). After the end of the experiments, femur and tibia were removed and analyzed for bone mineral density (BMD) by dual energy X-ray absorptiometry (DXA) and bone histomorphometry.

**Figure 2 biomedicines-08-00369-f002:**
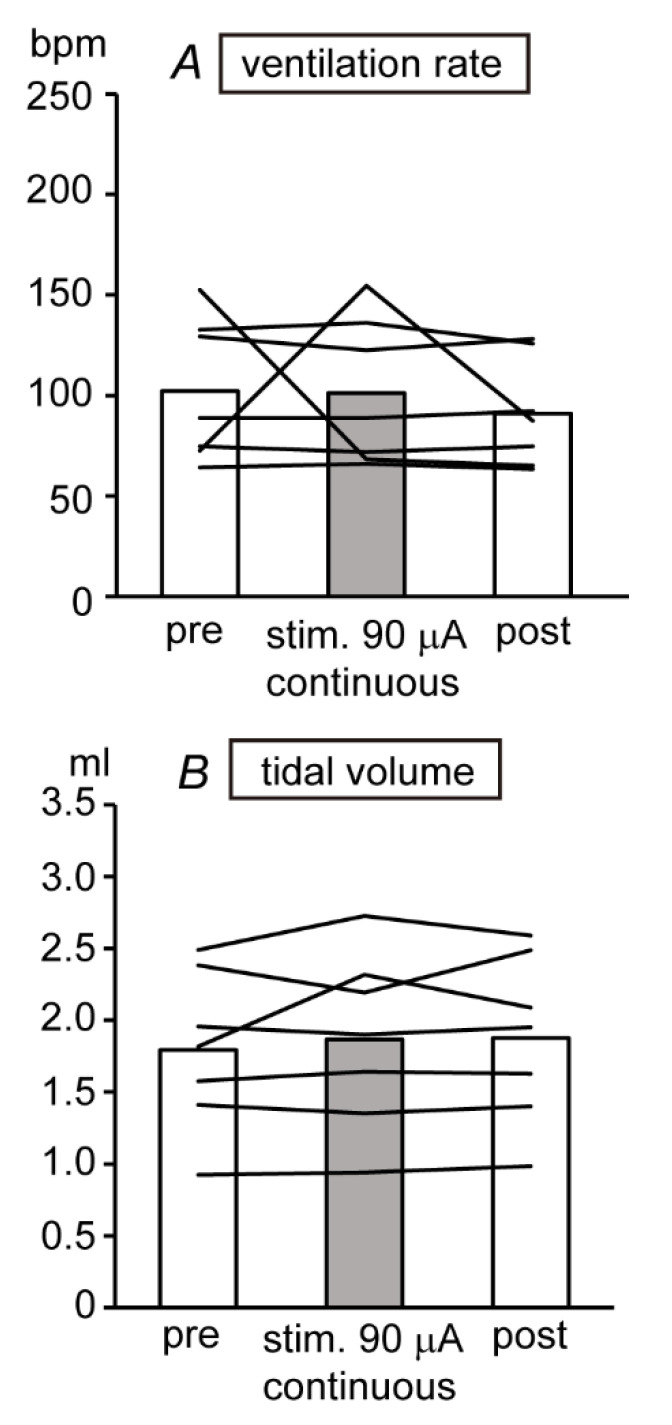
Effect of SLN stimulation (0.5 ms, 90 μA, continuous 40 Hz) on ventilation rate (**A**) and tidal volume (**B**) of conscious rats (*n* = 7). The ventilation rate and tidal volume were averaged 30 s or 5 min before, during, and after the stimulation. Each column represents mean value and lines indicate data obtained from individual rat.

**Figure 3 biomedicines-08-00369-f003:**
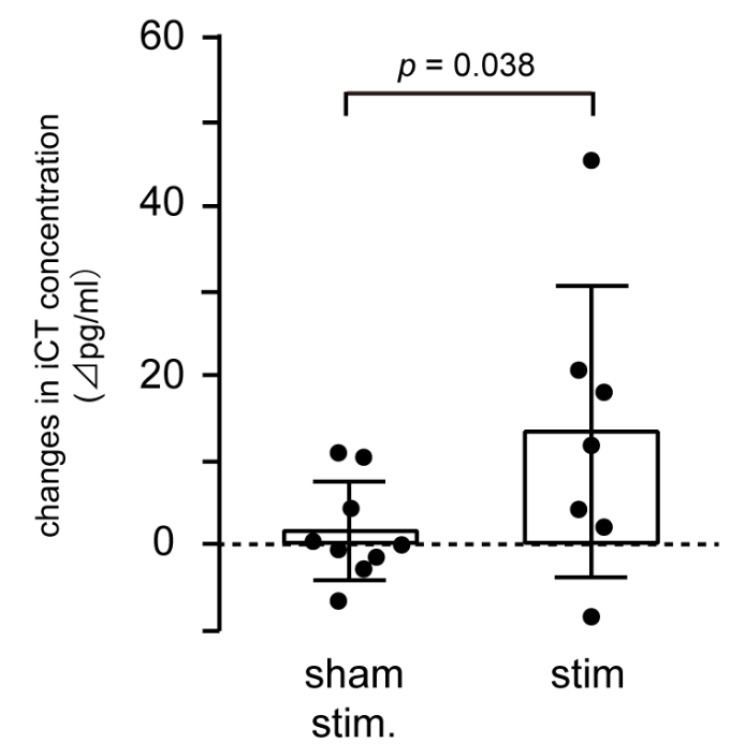
SLN stimulation increased systemic iCT concentration. Changes in systemic iCT concentration during stimulation (0.5 ms, 90 μA, continuous 40 Hz, 8 min) were compared between sham (*n* = 9 in three rats) and SLN (*n* = 7 in seven rats) stimulation. The Y-axis shows changes in iCT concentration from prestimulation values. A column represents mean value and closed circles indicate individual data. A *p* value indicates a significant difference (*p* < 0.05) tested by unpaired t-test (one-tailed).

**Figure 4 biomedicines-08-00369-f004:**
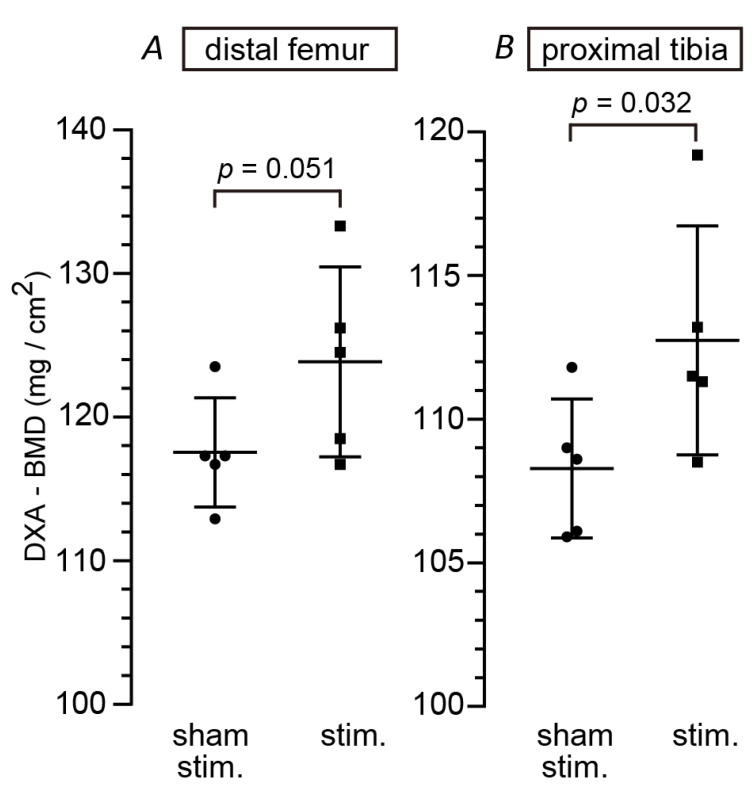
SLN stimulation increased BMD in OVX rats. After 3–4 weeks of SLN stimulation (0.5 ms, 90 μA, 40 Hz for 8 min/hour × 7 times/day × 4 days/week), femur and tibia were removed and BMD was measured by the DXA method. BMDs in distal metaphysis of femur (**A**) and proximal metaphysis of tibia (**B**) were compared between SLN-stimulated rats (*n* = 5) and nonstimulated (sham) rats (*n* = 5). Group data are expressed as mean ± SD. Closed circles and squares indicate data obtained from individual rats. The *p* values were obtained by unpaired *t*-test (one-tailed).

**Table 1 biomedicines-08-00369-t001:** Summary of the two experimental protocols.

Experiment	Gender	Gonadectomy	Measurements	SLN Stimulation *	Electrode Functionality *
Experiment 1	Male	Non	Respiration	Acute (7) †	Bilateral (5)
(16 rats)					Unilateral (2)
			Systemic iCT	Acute (7)	Bilateral (5)
					Unilateral (2)
				Sham (3)	Non
Experiment 2	Female	OVX	Bone analysis	Chronic (5)	Bilateral (3)
(10 rats)					Unilateral (2)
				Sham (5)	Non ^§^

* Numbers in brackets represent the number of rats used. † One rat was used for systemic immunoreactive calcitonin (iCT) measurement experiment also. ^§^ Electrodes were not implanted in two rats (head socket only).

**Table 2 biomedicines-08-00369-t002:** Effect of SLN stimulation in OVX rats on bone histomorphometry.

		OVX (*n* = 10)				
	Unit	Sham Stim. (*n* = 5)		SLN Stim. (*n* = 5)		*p* Value ^a^
Body weight	g	330	(320–349)	326	(316–344)	0.75
**Bone structure**						
Cancellous bone volume	%	8.8	(7.5–11.4)	8.4	(7.6–10.9)	0.94
Trabecular thickness	μm	61.2	(58.4–73.0)	67.5	(63.5–68.8)	0.31
Trabecular number	per mm	1.4	(1.1–1.8)	1.3	(1.1–1.6)	1.00
**Osteogenesis**						
Osteoid surface	%	8.1	(4.8–15.1)	8.3	(4.9–12.2)	1.00
Osteoblast surface	%	6.4	(3.0–7.7)	4.6	(4.4–5.8)	1.00
Bone calcification surface	%	26.8	(20.2–34.3) ^b^	30.6	(28.5–35.0)	0.56
Calcification rate	μm/day	1.3	(1.2–1.7) ^b^	1.6	(1.2–1.7)	0.81
**Bone resorption**						
Erosional surface	%	8.8	(5.7–11.6)	7.7	(5.9–9.1)	0.55
Osteoclast surface	%	3.7	(2.6–5.0)	4.2	(2.6–4.8)	0.84
Osteoclast number	per 100 mm	199	(141.6–305.3)	257	(189.6–317.5)	0.42

^a^ Mann–Whitney U-test, sham stimulation (stim.) vs. SLN stim. ^b^ Data from four rats. All values are expressed as the median and interquartile range (25%–75%).

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
