# Peer review of "Chronic Electrical Stimulation of the Superior Laryngeal Nerve in the Rat: A Potential Therapeutic Approach for Postmenopausal Osteoporosis"

_biomedicines, 2020, doi:10.3390/biomedicines8090369_

Round 1

Reviewer 1 Report

Dear Authors:

Your manuscript the title is “Chronic electrical stimulation of the SLN in the rat: apotential therapeutic approach for postmenopausal osteoporosis.”

This is a small article, but you want to publish in this journal SCI impact factor 4.717. I think need big.However, this is original not publish any else. Electrical stimulation is like acupuncture, the effects on the rat’s postmenopausal osteoporosis. I search the other publication in this journal, I find this journal no need too many results. Thus, I can agree you accepted to publication. However, still more figures to need.

Acupuncture is a traditional Chinese therapeutic approach. Is this the similar approach?

Forma: Introduction,Material and methods, Results, Conclusions.Like the other publications in this journal.

OVX is postmenopausal rats. This is the ovariectomized disease model of osteoporosis. Similar female woman stops estrogen induced osteoporosis.Using electrical stimulation to recur. Not else.

You used OVX (n = 2) and sham OVX (n = 2) rats to confirm development of osteoporosis following ovariectomy (OVX) in this experimental conditions, including recovery from head surgery and neck incision.N=2 is too small, need lots of, on the other hand, this is different from material and method expressed.

Superior laryngeal nerve (SLN)stimulation (0.5 ms, 90 uA,40 Hz)can induce calcitonin (CT)to therapy osteoporosis. Could you show me the stimulation positions as Figure 1?

Thank you       

Author Response

  • Acupuncture is a traditional Chinese therapeutic approach. Is this the similar approach?
  1. Thank you for your valuable comments. Our approach may have similarity to acupuncture in view of neuromodulation. Acupuncture therapy is less invasive neuromodulation and reportedly prevents the osteopenia induced by ovariectomy. In our approach, an invasive electrode implantation is necessary for direct electrical stimulation of the peripheral Superior Laryngeal Nerve. This enables targeted modulation of this nerve via electrical stimulation that is very selective. Differing electrical stimulation paradigms (e.g. frequency and amplitudes) can be administered directly to the nerve to alter functioning and achieve the desired result (increasing calcitonin release without adverse effects). As demonstrated in our experiments, electrodes were implanted successfully and used in conscious animals, where repeated timed stimulations could be applied to release calcitonin through the day. Such stimulation approaches could easily be controlled in a patient’s daily life to maximize the therapy.
  • Forma: Introduction,Material and methods, Results, Conclusions.Like the other publications in this journal.
  1. Thank you for your advice. We corrected the format in accordance with the instruction.
  • OVX is postmenopausal rats. This is the ovariectomized disease model of osteoporosis. Similar female woman stops estrogen induced osteoporosis. Using electrical stimulation to recur. Not else.
  1. Thank you for your comments.
  • You used OVX (n = 2) and sham OVX (n = 2) rats to confirm development of osteoporosis following ovariectomy (OVX) in this experimental conditions, including recovery from head surgery and neck incision. N=2 is too small, need lots of, on the other hand, this is different from material and method expressed.
  1. As the reviewer commented, “n = 2” is generally small. However, since an increase in body weight and a reduction in BMD following OVX have been well-established and characterized in the literature as an experimental model of osteoporosis, the number of animals used was minimized in this study due to the concordance of our measures with those reported within the literature. The methods for our preliminary experiments to confirm osteoporosis development was described in legend of supplementary Figure 1.

We modified a sentence of page 8, line 16 : “Consistent with previous reports [9, 10], in OVX rats, body weight was higher (Figure S1A), but BMD in the distal femur measured in vivo...”

  • Superior laryngeal nerve (SLN)stimulation (0.5 ms, 90 uA,40 Hz)can induce calcitonin (CT)to therapy osteoporosis. Could you show me the stimulation positions as Figure 1?
  1. The position where electrodes were implanted to stimulate the SLNs was the same as our previous study in anaesthetized rats. Thus, we added explanation on implantation position with a reference “(as shown in Fig. 1C of [5])” at the end of a sentence of page 4, line 5.

Reviewer 2 Report

Administration of calcitonin (CT) has been used to treat osteoporosis through suppressing bone resorbing osteoclasts. However, there is a problem of administration of CT due to its short half-life and antigenic activity. The authors group previously demonstrated that superior laryngeal nerve (SLN) stimulation innervates thyroid gland CT secretion. In this manuscript, Iimura et al. examined the effects of electrical stimulation of SLN on CT levels in blood. Additionally, they examined the effects of chronical nerve stimulation on bone mineral density using rat postmenopausal model (OVX). The authors revealed that stimulation of SLN increased blood CT levels than sham control without affecting spontaneous respiration. The authors also revealed that SLN stimulation increased BMD in OVX rat, despite SLN stimulation did not affect bone histomorphometry indices such as trabecular numbers, and osteoclast numbers. Based on these results, the authors proposed that SLN stimulation-induced increase in CT secretion is involved in inhibition of OVX-induced bone loss.

This manuscript is carefully written. Their conclusion is mostly supported by their experimental results. Discussion is well addressed to unsolved questions. I have just a few questions about their results.

SLN stimulation increased slightly but significantly BMD in proximal tibia, and nearly significantly in distal femur. However, the authors did not find any significant reduction of osteoclast numbers or bone resorbing activities, which should be inhibited by increased CT. They did not find increased osteogenesis in vivo, either. What do the authors explain this point?

Author Response

SLN stimulation increased slightly but significantly BMD in proximal tibia, and nearly significantly in distal femur. However, the authors did not find any significant reduction of osteoclast numbers or bone resorbing activities, which should be inhibited by increased CT. They did not find increased osteogenesis in vivo, either. What do the authors explain this point?

  1. Thank you for valuable comments. As the reviewer pointed out, we also wondered that SLN stimulation increased BMD, but did not affect bone histomorphometry.

It has been reported that BMD as well as examination of bone histomorphometry including bone resorption parameters are affected when a large amount of CT was pharmacologically (exogenously) administered. However, recent findings raised a possibility that the effect of CT differs between pharmacological administration (exogenic) and physiological secretion (endogenic) (JBMR 28(5):937-979, 2013). This matter was added in page 12, lines 1-6, as follows.

“In contrast to the slight but significant effect on BMD in proximal tibia, we did not find any significant reduction of osteoclast numbers or bone resorbing activities, nor increase in osteogenesis by histological analyses. Currently, our understanding of this unexpected result remains undetermined. One possibility is the difference may be too small to detect through histological examination. Alternatively, the chronic effect on bone cells of endogenous CT might be different from that of exogenous CT, as suggested by recent findings [27].”

Reviewer 3 Report

General comments

 In this study, they investigated the effects of chronic electrical stimulation of the superior laryngeal nerve (SLN)on serum calcitonin levels and probably trabecular bone mineral density (BMD) decreased by ovariectomy (OVX) in rats. They showed that SLN stimulation enhanced serum calcitonin levels in rats. Moreover, it significantly blunted the putative trabecular BMD at the tibia decreased by OVX).

 The data seemed to be interesting, the manuscript is well written. However, there are several issues, which should be addressed.

Specific comments

  1. It is generally known that the serum CT measurement is not reliable at the physiological level in human. The validation of CT assay and the related references should be described in the details in Methods.
  2. The regions of interest (ROIs) of BMD measurement is critical. Whether ROIs that they selected for BMD measurement was for the evaluation of trabecular bone should be clearly described following the generally accepted guideline (J Bone Miner Res 25:1468, 2010).
  3. Re:Page 4, lines 27-28, The description is not correct, since p value was > 0.05. The data was only tendency. The sentences should be revised. Moreover, the result differences between femoral and tibial bones should be discussed.
  4. Re: Figure 3, The data without OVX could be shown, if the authors had them. The data seemed to be preliminary.
  5. They could not show any data connecting calcitonin and BMD in this study. Therefore, this point should be described as the limitation of the study. Continuous calcitonin treatment induces downregulation of calcitonin receptor, then resulting in the disappearance of calcitonin effects. In human, calcitonin effects on bone is limited and already used for the treatment of osteoporosis, only for the decrease in bone pain through the neurological mechanism. This point should be discussed referring the effects of chronic electrical stimulation of the superior laryngeal nerve (SLN)of serum calcitonin levels in the present study.

Author Response

 It is generally known that the serum CT measurement is not reliable at the physiological level in human. The validation of CT assay and the related references should be described in the details in Methods.

  1. Thank you for the comments. In the present study, the concentration of immunoreactive CT was measured using ELISA. Thus, strictly speaking, we did not measure CT itself. Therefore, CT measured in the present study was amended to immunoreactive CT (iCT).

ELSA assay kits of rat CT used in the present study was thought to be reliable based on its sensitivity and precision. We have provide information on the assay kit in page 5, lines 13-14 as follows.

“using an ELISA kit (rat CT ELISA kit, MBS703165, MyBioSource, SanDiego, CA, minimum detectable dose: 0.39 pg/ml, intra-assay precision: CV% < 8%, inter-assay precision: CV% < 10%).”

3-2.    The regions of interest (ROIs) of BMD measurement is critical. Whether ROIs that they selected for BMD measurement was for the evaluation of trabecular bone should be clearly described following the generally accepted guideline (J Bone Miner Res 25:1468, 2010).      

  1. The position of ROI is critical as the reviewer mentioned. As described in the guideline, inter-individual difference in bone length may cause mislocation of ROIs. To avoid such an issue, we divided the whole bone into 20 parts of equal length on the long axis and placing 20 ROIs. Then, we confirmed the location of trabecular bone where BMD decreases severely following OVX (Figure S2), and used to determine effect of SLN stimulation.

The guideline which the reviewer kindly introduced was cited and amended the text in page 6, lines 29-30 as follows.

“The whole bone was divided into 20 parts of equal length to quantify each regional value to eliminate the influence of inter-individual difference in bone length [13].”

3-3.     Re:Page 4, lines 27-28, The description is not correct, since p value was > 0.05. The data was only tendency. The sentences should be revised. Moreover, the result differences between femoral and tibial bones should be discussed.

  1. In accordance with the reviewer’s advice, the descriptions have been amended.

Page 8, lines 33-34: “The BMD in the distal femur of the stimulation group tended to be higher than that in the sham group”

Page 11, lines 21-23: “The effect was statistically significant in proximal tibia and nearly significant in distal femur, but was similar in trabecular bone of both tibia and femur.

3-4.       Re: Figure 3, The data without OVX could be shown, if the authors had them. The data seemed to be preliminary.

  1. We appreciate the reviewer’s constructive comments. However, BMD values of sham OVX rats (i.e. rats that underwent surgery but did not have ovaries removed) is shown in both Figure S2 and Table S1 in supplementary materials. Therefore, readers can access these values obtained by preliminary experiments.

3-5-1.     They could not show any data connecting calcitonin and BMD in this study. Therefore, this point should be described as the limitation of the study.

  1. We appreciate the reviewer’s important advice and agree with the reviewer. We have added a sentence in page 11, line 37-38 as follows.

However, limitation of this study is that we could not show any data directly connecting CT and BMD.”

3-5-2.    Continuous calcitonin treatment induces downregulation of calcitonin receptor, then resulting in the disappearance of calcitonin effects. In human, calcitonin effects on bone is limited and already used for the treatment of osteoporosis, only for the decrease in bone pain through the neurological mechanism. This point should be discussed referring the effects of chronic electrical stimulation of the superior laryngeal nerve (SLN)of serum calcitonin levels in the present study.

  1. We appreciate the reviewer’s valuable comments. Indeed, the effect of CT administration on bone is limited in clinical setting and used for reducing bone pain. We have added this point and discussed in page 12, lines 6-11, as follows.

“…exogenous CT, as suggested by recent findings [27]. Salmon CT induces downregulation of CT receptor, resulting in the disappearance of CT effects. Consequently, exogenous CT’s effects on bone is limited in humans, now being used for the treatment of osteoporosis only for the decrease in bone pain via the neurological mechanism [3,28]. However, our initial results with neuromodulation of the SLN suggest increased endogenous CT secretion might operate through mechanisms differing from exogeneous CT for treatment of osteoporosis.”

Round 2

Reviewer 1 Report

I agree and accept this revised manuscript publish

Reviewer 2 Report

The authors replied to my question by adding additional discussion. It sounds reasonable. I'm satisfied the revised manuscript.